# Effect of Nano-Zirconia Addition on Mechanical Properties of Metakaolin-Based Geopolymer

**Muhammad Saukani [1], Ayu Novia Lisdawati [2], Heri Irawan [1], Rendy Muhamad Iqbal [3], Dwi Marta Nurjaya [4] and Sotya Astutiningsih [4,***

[1] Departement of Mechanical Engineering, Faculty of Engineering, Universitas Islam Kalimantan Muhammad Arsyad Al Banjari, Banjarmasin 70123, Indonesia
[2] Departement of Electrical Engineering, Faculty of Engineering, Universitas Islam Kalimantan Muhammad Arsyad Al Banjari, Banjarmasin 70123, Indonesia
[3] Department of Chemistry, Faculty of Mathematic and Natural Science, Universitas Palangka Raya, Kampus UPR Tunjung Nyaho, Palangka Raya 73111, Indonesia
[4] Department of Metallurgical and Materials Engineering, Faculty of Engineering, Universitas Indonesia, Depok 16424, Indonesia
* Correspondence: sotya.astutiningsih@ui.ac.id

**Abstract:** Geopolymer is an emerging material alternative to Portland cement and has potential as a refractory material. Adding filler in geopolymer material is a strategy to increase the advantages of its physical and mechanical properties. It has been previously reported that adding nanoparticles can increase the compressive strength value, but there is no reported interaction between nanoparticles and geopolymer during the geopolymerization process. This study aims to study the effect of adding nano-zirconia fillers on the physical and mechanical changes of metakaolin-based geopolymers with nano-zirconia fillers. The geopolymer samples were made with 100 g of metakaolin as the base material and nano-zirconia in amounts of 2 g, 5 g, 10 g, and 15 g. Further characterization was carried out by XRD, FTIR, and SEM-EDX. This study showed that the compressive strength of the MZr05 sample increased significantly by 58.7% compared to the control sample. The test results of the structure and functional groups did not lead to any new compounds formed in the geopolymerization reaction. Therefore, the interaction of metakaolin geopolymer with nano-zirconia only creates an interfacial bonding.

**Keywords:** compressive strength; metakaolin; nanomaterial; refractory; zirconia

## 1. Introduction

Geopolymer is one of the materials that has become the center of attention for researchers because it has many advantages in terms of physical, mechanical, and thermal properties. The compressive strength of geopolymers is superior to portland cement, even when heat treated above 400 °C [1]. The reaction involves the chemical reaction of various aluminosilicate oxides (in this case, $Al^{3+}$ with IV-fold coordination) with silicate solutions under high alkaline conditions. It produces Si–O–Al–O (silicon-oxo-aluminate) polymeric bonds [2,3]. Classified as inorganic polymers, geopolymers based on the silica/alumina ratio are classified into three groups, namely $SiO_2/Al_2O_3$ = 2, poly(sialate) (–Si–O–Al–O–), $SiO_2/Al_2O_3$ = 4, poly(sialate-siloxo) (–Si–O–Al–O–Si–O–), $SiO_2/Al_2O_3$ = 6, poly(sialate-disiloxo) (–Si–O–Al–O–Si–O–Si–O–) [4]. This material consists of an amorphous network of $AlO_4$ and $SiO_4$ tetrahedra connected by oxygen sharing. The presence of positive ions, such as $Na^+$, $K^+$, $Li^+$, and $Ca^{2+}$, is required to balance the negative charge of the IV-fold coordination of $Al^{3+}$ with oxygen [5].

The properties of geopolymer depend on the properties of the precursor. These properties include fineness, particle distribution, chemical composition, and reactive content of the geopolymer precursor. The aluminosilicate sources as basic materials used are

metakaolin, coal fly ash, and blast furnace slag [6–9]. The coal fly ash and blast furnace slag are cheap and abundant industrial wastes, but they have varying chemical and physical properties. Mainly they are used for substitution of ordinary Portland cement. In contrast to kaolin, which has consistent chemical properties and composition, it requires a dihydroxylation/amorphization process to activate its pozzolanic properties by forming kaolin [10]. The required calcination temperature varies between 650–850 °C with a holding time of 2–6 h [10–12].

Metakaolin-based geopolymer has the potential to be applied for encapsulation/immobilization of nuclear waste [13], bacterial-based self-healing [14], fast hardening after being given an activator, and reliable mechanical properties at room temperature [15–17]. However, metakaolin-based geopolymers have poor performance for high-temperature applications [6]. Therefore, it is necessary to add additives for modification to improve one or more geopolymer properties through chemical reactions and physical changes [18]. The additives can be organic or inorganic materials, solid or liquid, micro or nanoparticles, or fibers. It has been reported that several inorganic materials used as fillers to improve geopolymer performance include graphite, micro silica, nano silica, micro $TiO_2$, nano $TiO_2$, zircon, and micro zirconia [16,19–21].

Zirconia is an oxide material with high density and hardness, high thermal stability, high toughness, inertness, and anti-corrosion, so it is widely applied for structural and functional purposes [18,22]. This material is rarely found as $ZrO_2$ but naturally exists as Zircon ($ZrSiO_4$), better known as zircon sand. $ZrSiO_4$ can be processed into micro zircon, nano-zircon, silica, micro zirconia, and nano-zirconia [18,22–24].

The addition of zircon as a metakaolin-based geopolymer filler was reported to increase the compressive strength value by 23% [18]. Micro zirconia, as much as 3% as filler in fly ash-based geopolymer, increases compressive strength by 31% [23]. So far, the effect of nano-zirconia on the physical, chemical, and mechanical properties of geopolymer has not been reported. The addition of nanomaterials can produce ductile geopolymers with high compressive and tensile strengths [19]. It is because of the surface area to volume ratio of nanomaterial bulk material that will have different properties from its bulk state. Therefore, this paper assesses the effect of adding nano-zirconia fillers to change in physical and mechanical properties of metakaolin-based geopolymer.

## 2. Materials and Methods

### 2.1. Materials

Commercial metakaolin (Metastar 501) was purchased from IMERYS, the UK, with the mean particle size of 3 μm and the content of $SiO_2$ and $Al_2O_3$ is 56.0% and 38.1%, respectively [15], the microstructure and XRD pattern of which showed in Figure 1a,b with 12.93% degree of crystallinity (degree of crystallinity determined by Equation (3)). The moisture of metakaolin was reduced at 60 °C for 24 h. Nano-zirconia as filler was purchased from Hebei Souyi New Material Technology Co., Ltd., Handan, China, with an average particle size of 97 nm, the density of 5.88 g/cm$^3$, the microstructure and XRD pattern shown in Figure 1c,d. Sodium hydroxide pellets and sodium silicate solution were purchased from Merck, Germany, and Brataco Chemical Supply, Indonesia.

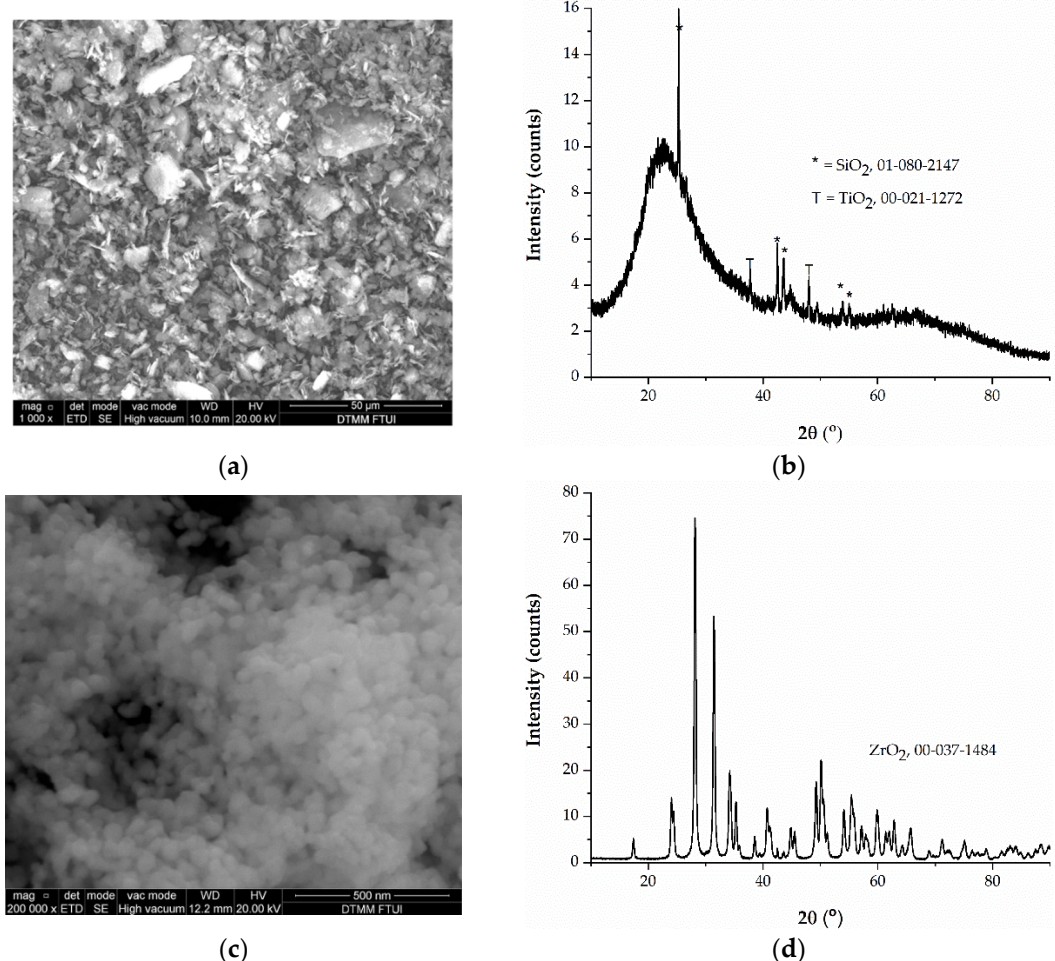

**Figure 1.** The microstructure and diffractogram of raw material (**a**) microstructure of metakaolin, (**b**) The diffractogram of metakaolin (* = SiO$_2$, T = TiO$_2$), (**c**) the microstructure of nano-zirconia, (**d**) diffractogram of zirconia.

*2.2. Methods*

2.2.1. Specimen Preparation

Design and sample preparation for geopolymer were similar to the previously reported by Zawrah et al. (2018) [18], with modifications on the molar ratio of Si/Al is 3.5, while Na$_2$O/SiO$_2$ and H$_2$O/SiO$_2$ ratios are as shown in Table 1. Geopolymer paste is differentiated into two categories: control and treated samples. The control sample is a sample with 100 g metakaolin as the precursor, while other samples are metakaolin with nano-zirconia addition (2 g (MZr02), 5 g (MZr05), 10 g (MZr10), and 15 g (MZr15).

**Table 1.** Mix design of geopolymer pastes.

| Sample | Metakaolin, g | Nano-Zirconia, g | Na$_2$O/SiO$_2$ | H$_2$O/Na$_2$O |
|---------|---------------|------------------|-----------------|----------------|
| Control | 100 | 0 | 0.351 | 8.933 |
| MZr02 | 100 | 2 | 0.354 | 9.269 |
| MZr05 | 100 | 5 | 0.359 | 9.759 |
| MZr10 | 100 | 10 | 0.367 | 10.538 |
| MZr15 | 100 | 15 | 0.375 | 11.274 |

Initially, metakaolin and nano-zirconia are mixed with a stand mixer for 3 h. The alkali solution is prepared by mixing 5 M NaOH with NaSiO$_4$. The amount of Na$_2$SiO$_4$ considered to obtain the molar ratio Si/Al on geopolymer paste is 3.5. The mixed material

was then slowly added alkali solution to form the homogeneity of geopolymer pastes. The paste was molded in a mortar mold with a volume of $5 \times 5 \times 5$ cm$^3$, then vibrated for 5 min, and the surface was covered with acrylic to decrease evaporation which caused the water loss. After 24 h at room temperature, samples were removed from the mold, kept at 80 °C for 24 h, and cured at room temperature for 28 days. Finally, the samples were heated at 100 °C for 24 h to stop the geopolymerization reaction.

2.2.2. Testing and Analysis Method

The functional group of geopolymer was recorded by FTIR spectrometer (Bruker Alpha II). The phase content and degree of crystallinity for raw material and geopolymer samples characterized by Bruker X-ray diffractometer (D8 Advanced ECO) using CuK$\alpha$1 wavelength (1.54056 Å), 2θ range from 10°–90° and interval step 0.0195°. The microstructure and mapping element of geopolymer samples were characterized by SEM-EDX (JOEL JSM-6390A). The compressive strength of specimens with a curing age of 28 days was evaluated by Universal Testing Machine according to ASTM C109.

The bulk density (B, g/cm$^3$) and apparent porosity (P, %) of geopolymer were calculated according to ASTM C20-00 (2010) by Equations (1) and (2) [25]. All data for compressive strength, porosity, and density from three sample experiments (n = 3) were represented as mean value and standard deviation. The relationship of nano-zirconia addition to compressive strength was identified by variance (ANOVA) using GraphPad Prism 9, and a P value less than 0.05 were considered statistically significant. Quantitative analysis of XRD data used X'Pert HighScore Plus, which equipped PDF2 data.

$$B \; = \; \frac{D}{W - S} \tag{1}$$

$$P \; = \; \frac{W - D}{W - S} \times 100 \tag{2}$$

where *D* is the samples' dry weight, *S* is the suspended weight, and *W* is the saturated weight.

$$Crystallinity \; = \; \frac{Area\,of\,Crystalline\,Peaks}{Area\,of\,all\,peaks\,(Cystalline + Amorphous)} \times 100 \tag{3}$$

## 3. Results and Discussion

Nanomaterials have different properties than their bulk counterparts, more chemically reactive owing to the large surface area to volume ratio. It has been reported that adding 3% nano-zirconia in castable refractories has increased flexural strength by up to 50% [26]. For this reason, the effect of nano-zirconia with 2 g, 5 g, 10 g, and 15 g for every 100 g of metakaolin in an alkaline environment to form geopolymers was studied.

The effect of nano-zirconia addition of metakaolin geopolymer to phase development and degree of crystallinity was observed using XRD. Figure 2a shows diffractograms of samples with and without nano-zirconia addition. The control sample shows the crystalline phases of $SiO_2$ and $TiO_2$ with PDF numbers 01-080-2147 and 00-021-1272, respectively. The metakaolin used as the base material has a crystallinity value of 12.93%, while after being mixed with alkali solution, the effect of hump was observed between the angles of 20° and 40° 2θ, and the degree of crystallinity decreased to 8.06%. This phenomenon indicated that the presence of an amorphous silica phase due to geopolymerization reaction [27].

After adding nano-zirconia, only $ZrO_2$ with PDF number 00-037-1484 was detected as a new phase and the degree of crystallinity was increased (Figure 2b). The qualitative analysis of the crystalline phase did not show any compound formed other than $SiO_2$, $TiO_2$, and $ZrO_2$. This indicates that the nano-zirconia, although it has a larger surface area to volume ratio than micro zirconia, does not show chemically different effects. Thus, the interaction between geopolymer and nano-zirconia was merely physical.

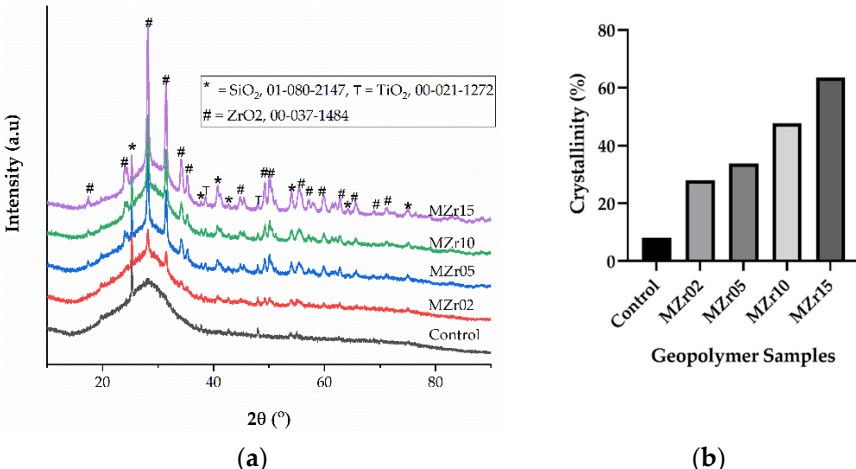

**Figure 2.** The XRD pattern of geopolymer samples with and without nano-zirconia addition (**a**,**b**) (* = $SiO_2$, T = $TiO_2$, and # = $ZrO_2$).

The spectra of FTIR are shown in Figure 3. The starting material, metakaolin spectra transmittance on wavenumber 1075 $cm^{-1}$ and 790 $cm^{-1}$ refer to asymmetric stretching vibration Al-O-Si of metakaolin and octahedral O-Al-OH functional groups [11,28], whereas the transmittance of zirconia in wavelength 756 $cm^{-1}$ and 667 $cm^{-1}$ are characteristic of monoclinic zirconia band [29]. The geopolymer samples showed similar spectra and the vibration around 976–991 $cm^{-1}$, representing Al-O-Si vibration. The shift of Al-O-Si vibration is owing to the amorphous formation from aluminosilicate gel [28]. The appearance of this vibration exhibits that the geopolymer network was successfully formed. Another vibration on 1650 $cm^{-1}$ represents H-O-H, it might be formed during the geopolymerization process due to the replacement position of counter ion from the source of base activator into geopolymer network, the $H_2O$ molecule only adsorbs as physical adsorption on the surface of geopolymer, it cannot form the chemical bond with the geo-polymer network. The vibration of Zr-O appears at around 690–700 $cm^{-1}$, there is no vibration which represents the formation of Zr-O-Si or Zr-O-Al in these spectra. Based on data, it can be suggested that there is no chemical interaction between $ZrO_2$ and the geopolymer network. It might be concluded that the formation of the composite can strengthen the physical properties of the geopolymer.

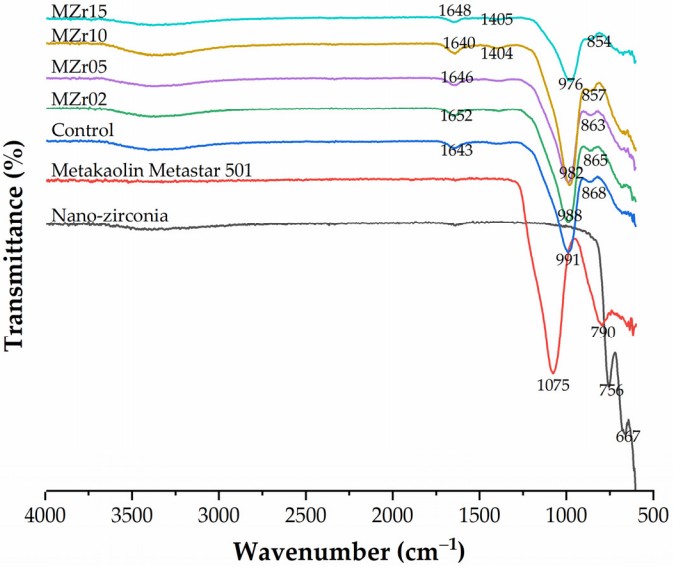

**Figure 3.** The FTIR spectra of starting materials and geopolymer samples.

The compressive strength of geopolymer was tested at the optimum strength development at the curing age of 28 days at room temperature, which is the same as for Ordinary Portland Cement-based material [7,14,17,30–33]. The same reference is applied to geopolymer composites [18,34,35]. Figure 4 shows the average compressive strength as an additional function of nano-zirconia with a molar Si/Al ratio of 3.5. The sample without adding nano-zirconia was used as a control sample compared to the effect of adding nano-zirconia to the compressive strength value. The control sample shows a compressive strength value of 33 MPa. Using the same metakaolin but different formulations of alkali activator resulted in different compressive strength values. Alkali activator with NaOH and water glass ratio 1:2 on metakaolin Metastar 501 only produces a compressive strength of 22.14 MPa [31]. The Si/Al molar ratio also influences compressive strength; it was reported that the Si/Al molar ratio of 2.2 resulted in a compressive strength value of 23 MPa [36].

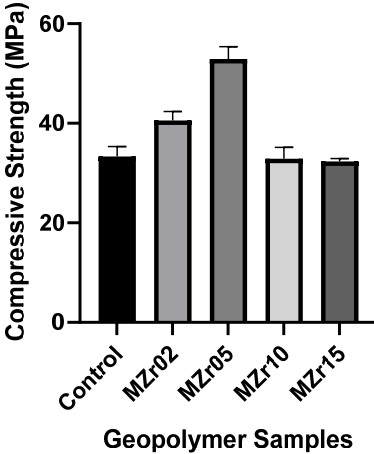

**Figure 4.** Compressive strength of geopolymer samples.

MZr02, MZr05, MZr10, and MZr15 showed compressive strength values of 40.59 MPa, 52.94 MPa, 32.89 MPa, and 32.33 MPa, respectively. The MZr05 sample showed a significant increase in compressive strength compared to the control sample, 58.75%, but MZr10 and MZr15 decreased the compressive strength value. The reduction of compressive strength is thought to be due to the addition of filler that exceeds the limit, inhibiting the reaction between aluminosilicate and alkali activator to form geopolymers [18].

Figure 5 shows morphology geopolymer samples with different amounts of nano-zirconia addition. Based on the XRD results in Figure 2, it is identified that the geopolymer samples are generally amorphous in shape, and there is unreacted metakaolin. The presence of unreacted metakaolin was due to the fast geopolymerization process. The unreacted metakaolin can cause structural defects so that it can reduce the compressive strength of geopolymer mortar [37,38]. The microstructure shown in Figure 5 indicates the microcrack triggered by the presence of unreacted metakaolin, which prevents a chemical reaction with alkali activator that leads to geopolymerization.

Figure 6 shows how the main elements of the sample are distributed, namely Al, Si, and Zr for samples MZr05 (a) and MZr15 (b). Green represents Al, Si is blue, and Zr is red. Based on the EDX results, the distribution of nano-zirconia in matrix geopolymer is almost homogenously distributed, although the existence of $ZrO_2$ piles of seen in some areas. Nevertheless, the ANOVA test's statistical tests showed that adding nano-zirconia as a filler in geopolymer composites had a significant effect with a p value of 0.0026 and $R^2$ = 0.934. Compared to the XRD pattern in Figure 2 and the FTIR spectra in Figure 3, it does not show any new phases and bonds between Zr-O-Si. This indicates that there is no chemical interaction between nano-zirconia and geopolymer but only physical interaction. Nano-zirconia functions as a filler in metakaolin-based geopolymer matrix that can precipitate and penetrate between the generated 3D network polysialate structures [18]. The decrease in compressive strength in samples MZr10 and MZr15 indicates that the limit amount of

nano-zirconia is 5 g for 100 g of metakaolin, where the excess filler can slacken the reaction of aluminosilicate and alkaline activator solution [7].

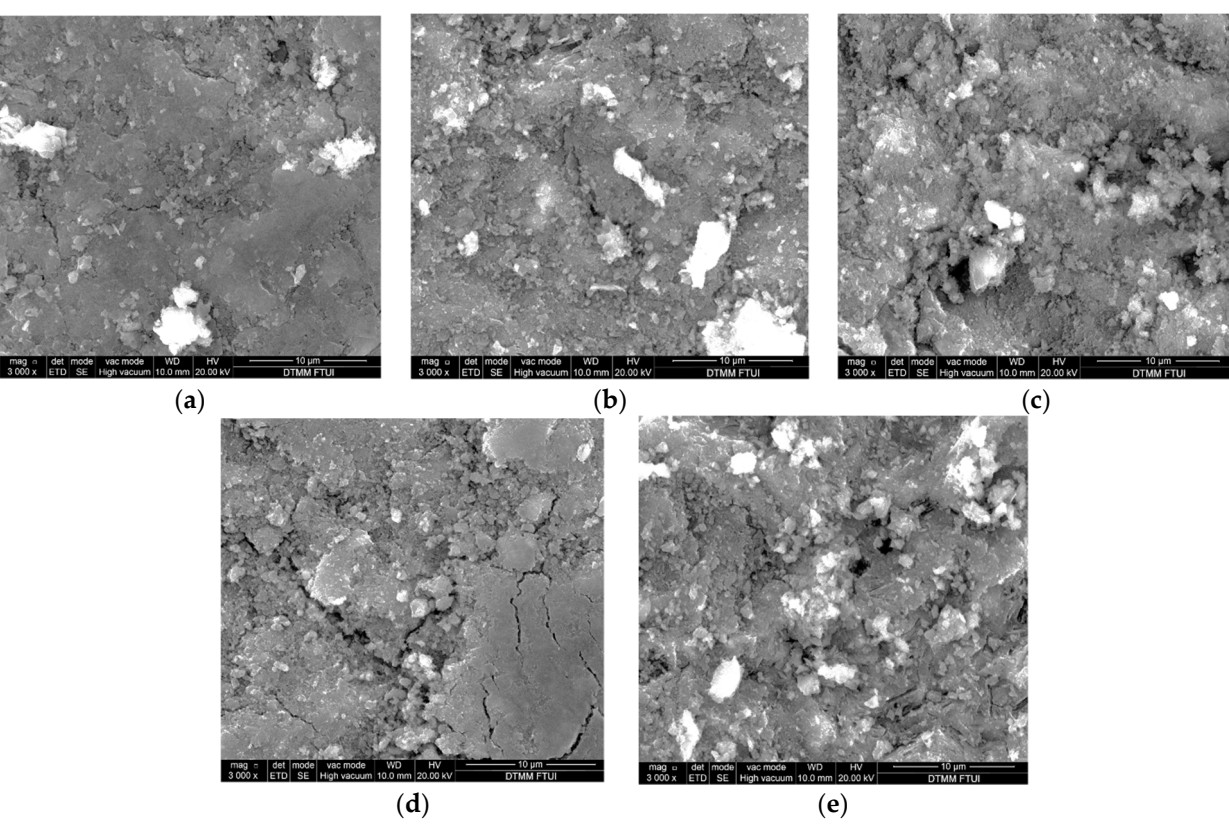

**Figure 5.** Microstructure of geopolymer sample, (**a**) Control, (**b**) MZr02, (**c**) MZr05, (**d**) MZr10, and (**e**) MZr15.

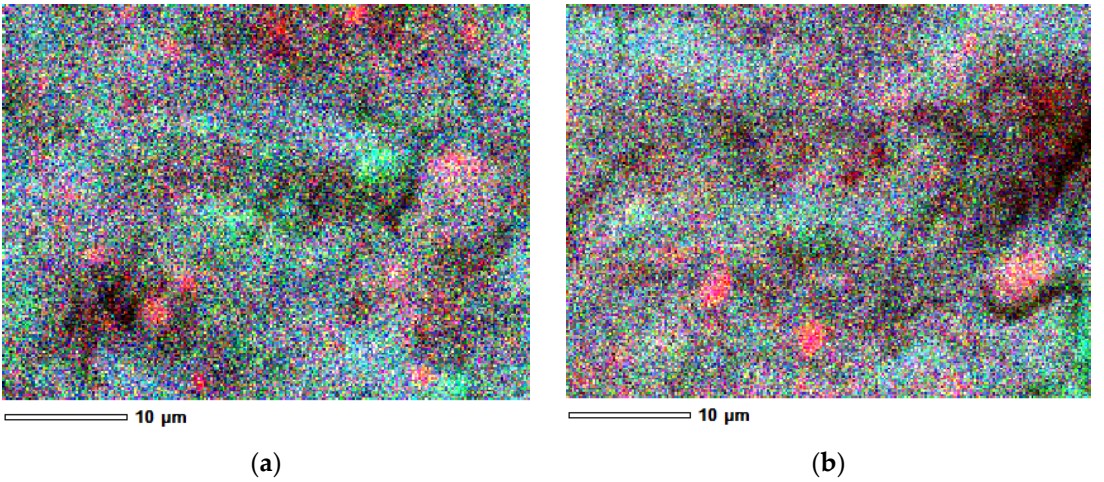

**Figure 6.** Scatter diagram of elements in geopolymer samples. (**a**) MZr05 and (**b**) MZr15.

The value of bulk density was obtained using the Archimedes method and calculated by Equation (1). The density of metakaolin is 2.62 g/cm$^3$, while the nano-zirconia density of nano-zirconia is 5.88 g/cm$^3$. Therefore, the increase in density will be directly proportional to the rise in the amount of zirconia contained in the sample. The resulting density value is 1.33–2.19 g/cm$^3$ (Figure 7). This value is greater than that of geopolymer with the addition of zircon, which is between 1.72–1.81 g/cm$^3$ [18].

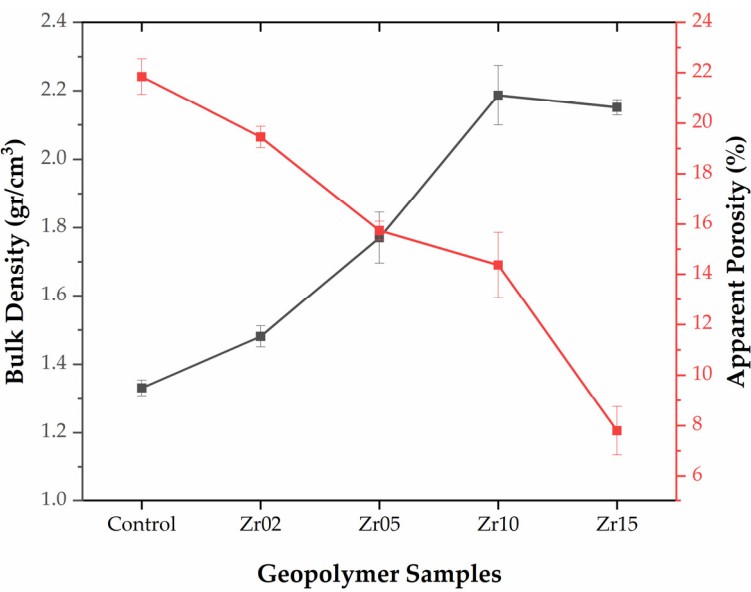

**Figure 7.** Bulk Density and apparent porosity of geopolymer samples.

### 4. Conclusions

Geopolymer metakaolin-based material has been prepared with the addition of nano-zirconia. XRD data showed that increasing content of nano-zirconia is in line with increasing degree of crystallinity specimens. Nano-zirconia addition improves compressive strength until 58.7% for 5 g nano-zirconia in 100 g metakaolin. The FTIR and XRD data revealed that interaction with nano-zirconia in geopolymer systems is only physical interaction.

**Author Contributions:** Conceptualization and validation, M.S., D.M.N., and S.A.; Methodology, M.S., R.M.I., and S.A.; Investigation, M.S., A.N.L., and H.I.; Resources, S.A.; Data curation, M.S. and R.M.I.; writing—original draft preparation, M.S., A.N.L., R.M.I.; writing—review and editing, S.A. and D.M.N.; visualization, H.I.; Supervision, S.A.; Project administration, A.N.L., and S.A.; Funding acquisition, M.S. and S.A. All authors have read and agreed to the published version of the manuscript.

**Funding:** This research was supported by grants from the Ministry of Education, Culture, Research, and Technology of Indonesia through the Research Collaboration Between Universities (PKPT) 2021 research scheme with the contract number: 60/LL11/KM/2021.

**Acknowledgments:** The authors would like to thank Musyarofah for the XRD and FTIR data acquisition at Kalimantan Institute of Technology (ITK).

**Conflicts of Interest:** The authors declare no conflict of interest.

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
