# Peer review of "Effect of Nano-Zirconia Addition on Mechanical Properties of Metakaolin-Based Geopolymer"

_jcs, doi:10.3390/jcs6100293_

Round 1
Reviewer 1 Report
The manuscript entitled "Effect of Nano-zirconia Addition on Mechanical Properties of Metakaolin-based Geopolymer" by Saukani et al. investigates the effect of adding nano-zirconia fillers on the physical and mechanical properties of metakaolin -based geopolymers.
Overall, it is a well structured study with recent references and well presented results that are supported by the figures.
I propose its publication after the revision of these three minor notes:
1) Please state the references for the equation (1) and (2),
2) line 220, by which equation? Eq. (1)?
3) Check English language in some sentences e.g. line 25-26, line 99 (there is no verb), lines 122-123.
Author Response
Response to Reviewer 1 Comments
Point 1: Please state the references for the equation (1) and (2)
Response 1: Thanks for your comment. The required referencing for equations (1) and (2) has been added.
Revision version:
The bulk density (B, g/cm3) and apparent porosity (P,%) of geopolymers were calculated according to ASTM C20-00 (2010) using equations (1) and (2) [25].
Reference:
- Shee-Ween, O.; Cheng-Yong, H.; Yun-Ming, L.; Abdullah, M.M.A.B.; Li Ngee, H.; Chan, L.W.L.; Wan-En, O.; Jaya, N.A.; Yong-Sing, N. Cold-Pressed Fly Ash Geopolymers: Effect of Formulation on Mechanical and Morphological Characteristics. Journal of Materials Research and Technology 2021, 15, 3028–3046, doi:10.1016/j.jmrt.2021.09.084.
Point 2: Line 220, by which equation? Eq. (1)?
Response 2: Thank you for your comment. In line 220, we have stated the equation to which we referred.
Revision Version:
The value of bulk density was obtained using the Archimedes method and calculated by equation (1).
Point 3: Check English language in some sentences e.g. line 25-26, line 99 (there is no verb), lines 122-123.
Response 3: Thanks for your comment and suggestion. We have checked and improved the english grammar for the sentences in line 25-26, line 99, and line 122-123.
Revision version:
Line 25 – 26 “The geopolymer samples were made of 100 g of metakaolin as the base material and nano-zirconia in amounts of 2 gr, 5 gr, 10 gr, and 15 gr.”
Line 99 – 100 “Design and sample preparation for geopolymer were similar to those previously reported by Zawrah et al. (2018) [18], with modifications on the molar ratio of Si/Al is 3.5, while Na2O/SiO2 and H2O/SiO2 ratios are as shown in Table 1.”
Line 122 – 123 “The bulk density (B, g/cm3) and apparent porosity (P,%) of geopolymers were calculated according to ASTM C20-00 (2010) using equations (1) and (2) [25]”
Referrence:
- Shee-Ween, O.; Cheng-Yong, H.; Yun-Ming, L.; Abdullah, M.M.A.B.; Li Ngee, H.; Chan, L.W.L.; Wan-En, O.; Jaya, N.A.; Yong-Sing, N. Cold-Pressed Fly Ash Geopolymers: Effect of Formulation on Mechanical and Morphological Characteristics. Journal of Materials Research and Technology 2021, 15, 3028–3046, doi:10.1016/j.jmrt.2021.09.084.

Reviewer 2 Report
In the present paper, the authors successfully developed high compressive strength metakaolin geopolymer paste. The results show that the compressive strength of the MZr05 sample increased significantly by 58.7% compared to the control sample.
View from the results of this work, this paper deserves the publication to Journal of Composites Science.
Author Response
Point 1: In the present paper, the authors successfully developed high compressive strength metakaolin geopolymer paste. The results show that the compressive strength of the MZr05 sample increased significantly by 58.7% compared to the control sample.
View from the results of this work, this paper deserves the publication to Journal of Composites Science.
Response 1: Thank you for your positive comment and your journal suggestion. We have asked the native speaker to proofread our article, and we have improved his suggestions.
